# AUTOREGRESSIVE PRETRAINING WITH MAMBA IN VISION

**Sucheng Ren**[1]  **Xianhang Li**[2]  **Haoqin Tu**[2]  **Feng Wang**[1]  **Fangxun Shu**[3]  **Lei Zhang**[4]
**Jieru Mei**[1]  **Linjie Yang**[5]  **Peng Wang**[5]  **Heng Wang**[5]  **Alan Yuille**[1]  **Cihang Xie**[2]

[1]Johns Hopkins University    [2]UC Santa Cruz    [3]Alibaba Group    [4]UCSD    [5]ByteDance

## ABSTRACT

The vision community has started to build with the recently developed state space model, Mamba, as the new backbone for a range of tasks. This paper shows that Mamba's visual capability can be significantly enhanced through autoregressive pretraining, a direction not previously explored. Efficiency-wise, the autoregressive nature can well capitalize on the Mamba's unidirectional recurrent structure, enabling faster overall training speed compared to other training strategies like mask modeling. Performance-wise, autoregressive pretraining equips the Mamba architecture with markedly higher accuracy over its supervised-trained counterparts and, more importantly, successfully unlocks its scaling potential to large and even huge model sizes. For example, with autoregressive pretraining, a base-size Mamba attains 83.2% ImageNet accuracy, outperforming its supervised counterpart by 2.0%; our huge-size Mamba, the largest Vision Mamba to date, attains 85.0% ImageNet accuracy (85.5% when finetuned with $384 \times 384$ inputs), notably surpassing all other Mamba variants in vision. The code is available at `https://github.com/OliverRensu/ARM`.

## 1  INTRODUCTION

In natural language processing (NLP), state space models (SSMs) (Gu et al., 2021a;b; Mehta et al., 2022; Gu et al., 2022) demonstrate strong potential for modeling long sequences with linear complexity. Among these, a recent variant, Mamba (Gu & Dao, 2023), has substantially advanced beyond traditional SSMs by synthesizing the best attributes of selective scanning. This innovation has also catalyzed its rapid adoption within the vision community, leading to its application across diverse visual tasks. These include the design of novel architectures (Liu et al., 2024b; Zhu et al., 2024; Huang et al., 2024; Pei et al., 2024; Wang et al., 2024a), applications to segmentation (Liu et al., 2024a; Wang et al., 2024b; Xing et al., 2024) and image synthesis (Guo et al., 2024).

However, these prior studies are mostly in the setting of supervised visual representation learning. While such trained models exhibit promising results in different visual tasks, they generally suffer from limited transferability and encounter notable difficulties in scaling He et al. (2022); Bao et al. (2022); He et al. (2020); Chen et al. (2020b). For example, as illustrated in Figure 1, attempts to scale the Vision Mamba (Vim) under supervised conditions often lead to either performance plateauing or even training collapse when pushed to very large sizes. These issues, therefore, motivate us to alternatively explore self-supervised visual representation learning with Mamba architectures, a method that has demonstrated notable successes in helping models secure strong and scalable visual representations (He et al., 2022; Bao et al., 2022; He et al., 2020; Chen et al., 2020b).

In this paper, we primarily focus on the autoregressive pretraining paradigm for self-supervised visual representation learning, which predicts the next token unidirectionally and autoregressively from the start to the end of the input sequence. This focus is driven by two reasons. First, autoregressive pretraining has already established itself as the de-facto standard in training large language models, with successful applications in various architectures including Transformers and Mamba (Dosovitskiy et al., 2020; Radford & Narasimhan, 2018; Gu & Dao, 2023). The recent literature has also successfully, albeit preliminarily, confirmed its efficacy in the computer vision domain, *e.g.*, helping Vision Transformer (ViT) develop strong and scalable feature representations (El-Nouby et al., 2024; Ren et al., 2023a). Secondly, Mamba architectures are inherently well-suited for autoregressive modeling

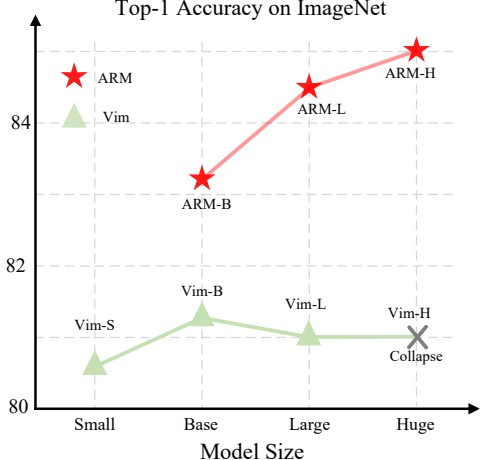

Figure 1: Compared to Vim, our ARM considerably boosts the ImageNet accuracy and, more critically, offers a stronger pathway for scaling up.

due to their uniquely designed linear attention nature, which methodically constructs token-wise relationships in a strictly progressive and unidirectional manner. This configuration ensures that each token can only attend to its preceding tokens, aligning perfectly with the underlying principles of autoregressive modeling. Additionally, this synergy practically leads to higher overall training efficiency. For example, under the setting of training the base-size Mamba for 300 epochs, autoregressive training requires only ~34 hours (measured by 8×A5000), a ~2× to ~10× improvement in training speed compared to other pretraining strategies (see Table 9 in Sec. 4.5).

Importantly, to further unleash the power of autoregressive visual representation learning with Mamba architectures, we highlight two key recipes for forming input sequences. First, instead of naively taking $16 \times 16$ patches as basic units of prediction, we opt for a more strategic approach by grouping spatially neighboring patches to form larger clusters; empirically, we find the cluster size of $64 \times 64$ reaches the best performance. Secondly, in our ablation of mapping 2D images into 1D visual sentences with various orderings, we note that vanilla ordering, which simply orders clusters with the row-by-row and forward scan approach, is already an effective choice. We term this method ARM.

Extensive results are provided showing our proposed ARM achieves substantially stronger performance. As shown in Figure 1, ARM helps our base-size model attain 83.2% ImageNet accuracy, outperforming the supervised counterpart by 2.0% and achieves 85.2% Top-1 accuracy with the input resolution of $448\times448$. Moreover, ARM enables the training of the first successful huge-size model (ARM-H), marking it as the largest vision Mamba model to date. Specifically, ARM-H achieves an impressive 85.0% ImageNet accuracy, significantly outperforming all previous Mamba variants. Additionally, ARM also improves the performance on out-of-domain datasets by a large margin: ARM-B outperforms supervised Vim-B by 4.4% on ImageNet-A, 2.9% on ImageNet-R, and 3.3% on ImageNet-S.

## 2 RELATED WORK

**State space model.** The state space model (SSM) (Gu et al., 2021a) stands as a novel alternative to Transformers for long-range dependency modeling with linear complexity. Linear attention (Katharopoulos et al., 2020; Choromanski et al., 2020; Peng et al., 2021) recurrently approximates self-attention via a softmax-free attention matrix with linear complexity, which can be viewed as a degenerate linear SSM. The Structured State-Space Sequence (S4) model (Gu et al., 2021a) computes more efficiently than prior approaches while preserving their theoretical strengths based on a new parameterization. S5 (Smith et al., 2022) extends S4 by adding multi-input multi-output (MIMO) SSM and efficient parallel scan. RWKV (Peng et al., 2023) is a recent RNN with its key "WKV" components that operate similarly to a system with two SSMs. Its updated version (Peng et al., 2024) incorporates state expansion and input-dependent gating for more flexible sequence modeling. Following this, Mamba (Gu & Dao, 2023) proposes a data-dependent SSM layer with

hidden state expansion and builds a generic language model backbone, which performs comparably to transformers at various sizes and enjoys linear scaling in sequence length. This work focuses on Mamba in vision, aiming to enhance it via autoregressive visual pretraining.

**Mamba in vision.** The successful application of Mamba in NLP has inspired its adoption in vision applications. Vision Mamba (Vim) (Zhu et al., 2024) utilizes Vim blocks composed of pure Mamba layers: each Vim block leverages both forward and backward scans to model bidirectional representations and mitigate the direction-sensitive problem in Mamba. Alternatively, Vmamba (Liu et al., 2024b) employs Visual State Space (VSS) blocks that integrate both Mamba and 2D convolution layers, supported by a pyramid architecture akin to the Swin Transformer (Liu et al., 2021): each VSS block first models 2D local information via 2D depth-wise convolution as the token mixer, followed by a CrossScan Module that processes 2D global information both horizontally and vertically. Mamba-ND (Li et al., 2024) further expands Mamba's capabilities to multi-dimensional data, including images and videos. LocalMamba (Huang et al., 2024) splits the input image into several local windows and performs SSM in various directions within these windows, enhancing local processing. EfficientVMamba (Pei et al., 2024) introduces an efficient 2D scanning technique using atrous sampling on feature map patches to reduce computational demands. Compared to these newly designed Mamba architectures, ours is *less novel*, which closely follows the design of ViT, but substituting the self-attention with the Mamba module. With this *naive* Mamba architecture, our main focus is to show autoregressive pretraining can enhance its visual capabilities.

**Self-supervised visual representation learning.** Self-supervised visual representation learning (Chen et al., 2020c; He et al., 2020; Chen et al., 2021; 2020b; Ren et al., 2022a; 2024; Zhai et al., 2022; He et al., 2022; Bao et al., 2022; Ren et al., 2023b) aims to learn strong and transferable representations without labels, including contrastive learning (Chen et al., 2020c; He et al., 2020; Chen et al., 2021; 2020b), position prediction (Zhai et al., 2022), masked image modeling (He et al., 2022; Bao et al., 2022; Ren et al., 2023b), *etc*. This paper focuses on autoregressive pretraining, which is highly successful in NLP but still less explored in computer vision. iGPT (Chen et al., 2020a) is the first work to introduce Generative Pretrained Transformer to vision and highlights the potential of autoregressive pretraining as a general self-supervised visual representation learning strategy. SAIM (Qi et al., 2023) and RandSAC (Hua et al., 2022) further enhance autoregressive pretraining, achieving performance on par with MAE (He et al., 2022) by utilizing the ViT architecture and a stochastic sequence permutation strategy. D-iGPT (Ren et al., 2023a) slightly modifies the learning objective to predict not only the next token but also visible tokens. AIM (El-Nouby et al., 2024) demonstrates that, with autoregressive pretraining, ViT scales effectively with increased model capacity and data quantity. Different from these prior works, which focus on Transformer architecture, we provide the first study of exploring autoregressive visual pretraining with Mamba architectures.

## 3 METHOD

### 3.1 MAMBA PRELIMINARIES

The Mamba architecture inherits from state space sequence models (Gu et al., 2021a), which models a 1-D function or sequence $x(t) \in \mathbb{R} \to y(t) \in \mathbb{R}$ at time $t$ via expanded hidden states $h_t \in \mathbb{R}^N$. The hidden state is evolved through time driven by parameters $\mathbf{A}, \mathbf{B}, \mathbf{C}$ following linear ordinary differential equations (ODEs):

$$\begin{aligned} h'(t) &= \mathbf{A}h(t) + \mathbf{B}x(t), \\ y(t) &= \mathbf{C}h(t). \end{aligned} \tag{1}$$

To discretize parameters in this continuous system, a common solution is to introduce a time scale parameter $\boldsymbol{\Delta}$ to transform continuous $\mathbf{A}, \mathbf{B}$ to discrete $\overline{\mathbf{A}}, \overline{\mathbf{B}}$ using zero-order hold (ZOH) model (Oppenheim et al., 1997):

$$\begin{aligned} \overline{\mathbf{A}} &= \exp(\boldsymbol{\Delta}\mathbf{A}), \\ \overline{\mathbf{B}} &= (\boldsymbol{\Delta}\mathbf{A})^{-1}(\exp(\boldsymbol{\Delta}\mathbf{A}) - \mathbf{I}) \cdot \boldsymbol{\Delta}\mathbf{B}. \end{aligned} \tag{2}$$

By applying such transformation, we can rewrite Eq. 1 as:

$$\begin{aligned} h'_t &= \overline{\mathbf{A}}h_{t-1} + \overline{\mathbf{B}}x_t, \\ y_t &= \mathbf{C}h_t. \end{aligned} \tag{3}$$

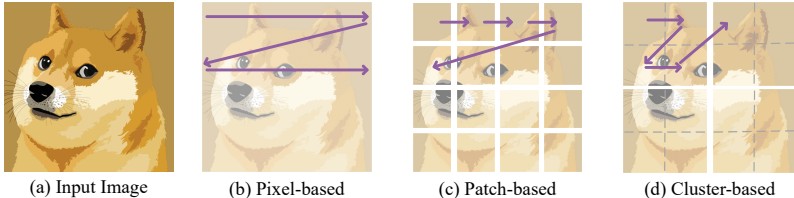

| (a) Input Image | (b) Pixel-based | (c) Patch-based | (d) Cluster-based |

Figure 2: Different prediction units in the autoregressive modeling.

We then employ a matrix $\overline{\mathbf{K}}$ for fast computation:

$$\overline{\mathbf{K}} = (\mathbf{C}\overline{\mathbf{B}}, \mathbf{C}\overline{\mathbf{A}}\overline{\mathbf{B}}, ..., \mathbf{C}\overline{\mathbf{A}}^k\overline{\mathbf{B}}, ...),$$
$$\mathbf{y} = \mathbf{x} * \overline{\mathbf{K}}, \tag{4}$$

where $k \in [0, L)$ and $L$ is the input sequence length. We also have $\mathbf{y} = \{y_1, ..., y_L\}$, $\mathbf{x} = \{x_1, ..., x_L\}$, while $\overline{\mathbf{K}} \in \mathbb{R}^L$ can be regarded as the convolutional kernel. Note this computing structure allows Mamba to model the input sequence that perfectly matches the unidirectional, next-word prediction in autoregressive modeling.

## 3.2 Autoregressive pretraining

We first briefly revisit autoregressive pretraining in NLP. Then, we shift our attention to autoregressive pretraining with mamba in vision, including the prediction unit and prediction order design. Lastly, we present the model variants.

### 3.2.1 Autoregressive Pretraining in NLP

Autoregressive pretraining models the probability of the next word one by one given a corpus $\mathcal{U} = \{u_1, ..., u_n\}$. This can be formulated as:

$$p(u) = \prod_{i=1}^{n} p(u_i | u_1, ..., u_{i-1}, \Theta) \tag{5}$$

Here, autoregressive pertaining computes the likelihood of each word $u_i$ based on the context of all preceding words from $u_1$ to $u_{i-1}$ and minimizes the negative log-likelihood:

$$\mathcal{L} = -log \ p(u) \tag{6}$$

This strategy plays a fundamental role in training large language models like ChatGPT (Brown et al., 2020) and GPT-4 (OpenAI, 2023) in NLP.

### 3.2.2 Autoregressive Pretraining with Mamba in Vision

**Prediction unit.** Transitioning from 1D sentences to 2D images introduces the challenge of defining a suitable autoregressive prediction unit. We start with the vanilla strategy presented in iGPT (Chen et al., 2020a) which considers each individual pixel as the prediction unit, as illustrated in Figure 2(b). For an image $X = \{p_1, ..., p_n\}$, our objective is to minimize the loss function:

$$\mathcal{L} = \sum_{i=1}^{n-1} l(f([p_1, ..., p_i]), p_{i+1}),$$
$$l(\hat{y}, y) = |\hat{y} - y|^2. \tag{7}$$

Here $f(\cdot)$ denotes the Mamba model, and $p_i$ represents the $i_{th}$ pixel of the image. This pixel-based approach, while straightforward, imposes significant computational demands, particularly for high-resolution images. Therefore, as shown in the original iGPT paper (Chen et al., 2020a), this constraint necessitates the use of low-resolution images for computationally feasible autoregressive pretraining.

Patchifying (Dosovitskiy et al., 2020) images into non-overlapped regions and then mapping them into visual tokens can address this computation challenge. For example, with an image size of

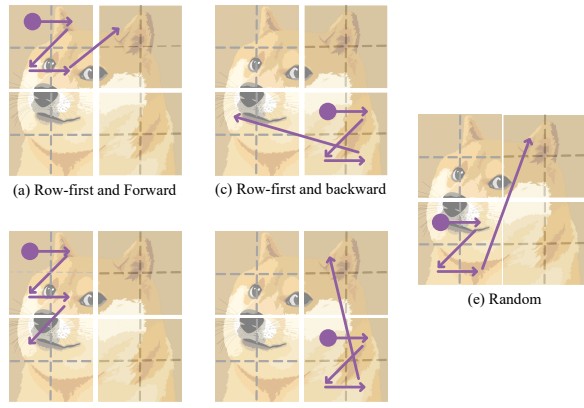

Figure 3: Different prediction orderings of a visual sentence.

$224 \times 224$, the sequence length would reduce significantly from 50,176 in the iGPT framework to just 196 patches with the $16 \times 16$ patchifying operation. Intuitively, shifting the prediction unit from pixels (Chen et al., 2020a) to patches (Dosovitskiy et al., 2020; Zhu et al., 2024; El-Nouby et al., 2024), as shown in Figure 2(c), adjusts the autoregressive input to $X = \{P_1, ..., P_n\}$:

$$\mathcal{L} = \sum_{i=1}^{n-1} l(f([P_1, ..., P_i]), P_{i+1}), \tag{8}$$

$$l(\hat{y}, y) = |\hat{y} - y|^2.$$

Here $P_i \in \mathcal{R}^{16 \times 16}$ is the $i_{th}$ patch. Moreover, to encapsulate the 2D spatial information at the token level, we propose grouping spatially adjacent patches into larger clusters to serve as the prediction unit, illustrated in Figure 2(d). The clustered input $X = \{c_1, ..., c_n\}$ aims to be optimized by:

$$\mathcal{L}_{\text{ARM}} = \sum_{i=1}^{n-1} l(f([c_1, ..., c_i]), c_{i+1}), \tag{9}$$

$$l(\hat{y}, y) = |\hat{y} - y|^2.$$

Here, each $c_i \in \mathcal{R}^{H_c \times W_c}$ is a cluster formed by grouping $\frac{H_c}{16} \times \frac{W_c}{16}$ patches, and our model scans cluster by cluster. In each cluster, our model scan patch by patch. The ablation studies (Section 4.5, Table 5) show that using clusters as prediction targets significantly enhances performance compared to the use of individual pixels or patches. Next, we explore the strategies for sequencing these clusters into a coherent visual sentence.

**Prediction order.** Unlike the 1D sentences in NLP, which inherently have a clear sequence order for autoregressive modeling, we hereby explore four different prediction orders when projecting 2D images into 1D visual sentences, *e.g.*, how these clusters should be arranged given a cluster size of $s$, with $\frac{W}{s}$ clusters per row and $\frac{H}{s}$ clusters per column. We hereby explore four primary prediction orders: 1) *Row-first and forward* orders the clusters row by row, processing from the first to the last cluster within each row sequentially, as depicted in Figure 3(a). 2) *Row-first and backward* similarly orders the clusters row by row but inverts the processing direction, starting with the last cluster and moving to the first within each row, illustrated in Figure 3(b). 3) *Column-first and forward* organizes the clusters column by column, processing sequentially within each column from top to bottom, shown in Figure 3(c). 4) *Column-first and backward* similarly sequences the clusters column by column but starts with the bottom-most cluster, moving upwards, as seen in Figure 3(c). To consider an approach free from pre-defined sequential biases, we also experimented with a *Random* permutation (Yang et al., 2019) of cluster order, visualized in Figure 3(e). Note this randomness is running on the fly, *i.e.*, the same image will be processed in different orders at different training steps. We add position embedding for all prediction order designs.

Detailed empirical comparisons of these four predefined orders alongside the random order are presented in Section 4.5. Our findings reveal that while the predefined orders exhibit minimal differences in performance, employing a random order leads to severe performance degradation. Consequently, the straightforward and effective *row-first and forward order* (Figure 3(a)) is adopted as our standard ordering strategy for autoregressive modeling.

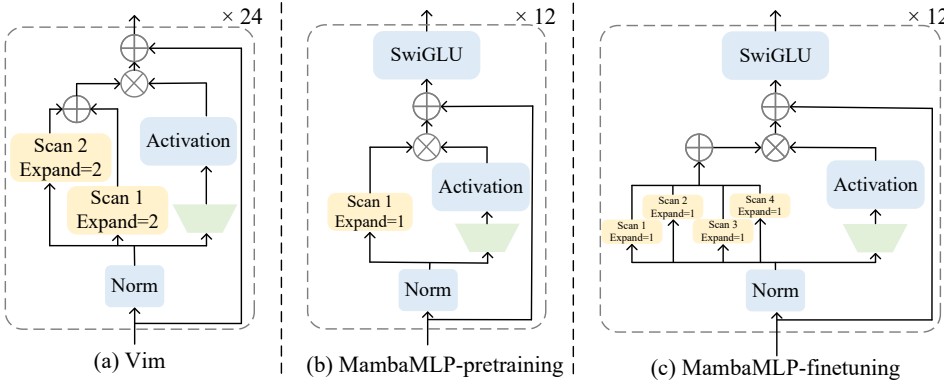

Figure 4: The comparison of block architectures between Vim, and MambaMLP in pretraining and in finetuning.

### 3.3 MAMBAMLP

We hereby introduce our newly developed MambaMLP blocks. Specifically, drawing inspiration from the self-attention block in Transformer (Dosovitskiy et al., 2020; Vaswani et al., 2017), our MambaMLP block uses Mamba as the token mixer and keep the multi-layer perceptron (MLP) the same as that in the self-attention block. Note that the configuration of the MambaMLP block varies between pretraining and fine-tuning phases to cater to their different requirements. During pretraining, as illustrated in Figure 4(b), the MambaMLP block contains the Mamba layer with only 1 scan (Liu et al., 2024b). We stress that using 1 scan is a hard requirement in pretraining — this is because autoregressive modeling requires a strict causal structure, where each token is predicted based only on past tokens in the defined sequence order; if multiple scans are used, this autoregressive nature will break and information leakage will happen, resulting in the learned feature representation collapsing. In finetuning (shown in Figure 4(c)), the block is then adapted to contains the Mamba layer with the standard 4 scans as in Vmamba (Liu et al., 2024b), which enables bi-directional modeling of global information to strengthen the overall performance. Note that in these 4 scans, the parameters are all initialized using the values of the parameters from the single scan during pretraining. The other architectural components in the pretraining and the finetuning stay the same: the block utilizes SwiGLU (Touvron et al., 2023) as the MLP layer, and the $expand$ refers to the order of the sequence that the Mamba processes and is set to 1 to enhance scanning efficiency. Additionally, we provide a visual comparison between our MambaMLP block and the Vim block in Figure 4. We can see that the Vim block contains Mamba layers with 2 scans (Liu et al., 2024b) for bi-directional global information processing and has no MLP layer, and the $expand$ of each scan is set to 2. Practically, this larger $expand$ in each scan results in higher performance but slower inference speeds.

By stacking multiple MambaMLP blocks and training with our autoregressive strategy developed in Section 3.2.2, we name the resulting model ARM. As detailed in Table 1, ARM is designed to match the depth and width of ViT in its base and large configurations. For the huge model size, ARM adopts the structure of AIM-600M (El-Nouby et al., 2024), which is wider but less deep compared to ViT-H, balancing performance and computational efficiency. The decoder follows exactly the same design as in MAE (He et al., 2022), which uses Transformer blocks (by default, we choose depth=4 and width=512). Note that the decoder is only used for pretraining and we remove the decoder in downstream tasks. In the next section, we will extensively validate the efficacy of ARM.

## 4 EXPERIMENT

### 4.1 IMPLEMENTATION DETAILS

**Pretraining.** We pretrain ARM using the ImageNet-1K dataset (Deng et al., 2009). Specifically, ARM-B and ARM-L are pre-trained for 1600 epochs, and ARM-H is pre-trained for 800 epochs. We use a batch size of 2048/1024/512 for ARM-B/L/H, respectively, and a learning rate of lr = 1.5e-4$\times\frac{\text{batchsize}}{256}$. We adopt a cosine decay schedule with a warm-up for 5 epochs. We adopt the AdamW (Loshchilov & Hutter, 2019) optimizer with a weight decay of 0.05. We use random resized cropping and random horizontal flipping. The pretraining input size is set to $192 \times 192$.

Table 1: The configuration of different architecture variants.

| Model | Block | Width | Depth | Param.(M) |
|-------|-------|-------|-------|-----------|
| ViT-B | (Attention+MLP) | 768 | 12 | 86 |
| Vim-B | Mamba | 768 | 24 | 98 |
| ARM-B | (Mamba+MLP) | 768 | 12 | 85 |
| ViT-L | (Attention+MLP) | 1024 | 24 | 307 |
| Vim-L | Mamba | 1024 | 48 | 340 |
| ARM-L | (Mamba+MLP) | 1024 | 24 | 297 |
| ViT-H | (Attention+MLP) | 1280 | 32 | 632 |
| Vim-H | Mamba | 1536 | 48 | 755 |
| ARM-H | (Mamba+MLP) | 1536 | 24 | 662 |

Table 2: Performance comparison on ImageNet-1K. Throughputs are measured with an A5000 GPU. † denotes we extend the training of Vim to the large-size model, using its original GitHub repo. ‡ indicates the stride is 8. Hybrid architectures are in Gray.

| Model | Token Mixer | Image Size | Param. (M) | Throughputs (imgs/s) | Top-1 (%) |
|-------|-------------|------------|------------|----------------------|-----------|
| *Base-size models* | | | | | |
| RegNetY-16G | 2D Conv. | $224^2$ | 84 | 870 | 82.9 |
| DeiT-B | Attention | $224^2$ | 21 | 1073 | 81.2 |
| Vim-B† | Mamba | $224^2$ | 98 | 890 | 81.2 |
| MambaMLP-B | Mamba | $224^2$ | 85 | 1301 | 81.2 |
| VMamba-B | Mamba+2D Conv. | $224^2$ | 89 | 315 | 83.9 |
| ARM-B | Mamba | $224^2$ | 85 | 1301 | 83.2 |
| ARM-B | Mamba | $384^2$ | 85 | 440 | 84.2 |
| ARM-B ‡ | Mamba | $448^2$ | 85 | 86 | 85.2 |
| *Large-size models* | | | | | |
| Vim-L† | Mamba | $224^2$ | 340 | 345 | 81.0 |
| MambaMLP | Mamba | $224^2$ | 297 | 445 | 81.4 |
| ARM-L | Mamba | $224^2$ | 297 | 445 | 84.5 |
| ARM-L | Mamba | $384^2$ | 297 | 154 | 85.1 |
| *Huge-size models* | | | | | |
| Vim-H† | Mamba | $224^2$ | 755 | 211 | collapsed |
| ARM-H | Mamba | $224^2$ | 662 | 275 | 85.0 |
| ARM-H | Mamba | $384^2$ | 662 | 94 | 85.5 |

**Finetuning.** Following pretraining, we finetune the ARM models on the ImageNet classification task. Specifically, we finetune all models for 100 epochs with a batch size of 1024, with the input size set at $224 \times 224$. We use the same data augmentation as MAE (He et al., 2022). We adopt AdamW as an optimizer, and the peak learning rate is lr=5e-4$\times\frac{\text{batchsize}}{256}$ with a cosine decay schedule and a warm-up for 5 epochs. Additionally, we employ the exponential moving average (EMA) (Izmailov et al., 2018) for stronger performance.

In addition to testing on the ImageNet evaluation set, we evaluate model robustness without finetuning on various out-of-domain ImageNet variants, including natural adversarial examples (ImageNet-A (Hendrycks et al., 2021b)), semantic shifts (ImageNet-R (Hendrycks et al., 2021a)), image sketches (ImageNet-S (Wang et al., 2019)), ImageNet-V2 (Recht et al., 2019), and ImageNet-Real (Beyer et al., 2020). Moreover, we finetune the pretrained model on different downstream tasks including object detection and instance segmentation on COCO (Lin et al., 2014), and semantic segmentation on ADE20K (Zhou et al., 2019).

## 4.2 MAIN RESULTS

In Table 2, we compare our ARM with convolution-based RegNet (Radosavovic et al., 2020), Attention-based ViT, and different Mamba architectures in vision. For the base-size model, our ARM achieves 83.2% accuracy, making a substantial 2.0% improvement over its supervised MambaMLP counterpart. Additionally, we note that ARM outperforms Vim by 2.0%, and is the only Mamba architecture that attains stronger performance than convolution-based RegNetY-16G (*i.e.*, by 0.3%).

Table 3: Robustness and Generalization evaluation on out-of-domain datasets.

| Method | IN-1K ↑ | IN-V2 ↑ | IN-Real ↑ | IN-Adv.↑ | IN-Ren.↑ | IN-Ske.↑ |
|---|---|---|---|---|---|---|
| Vim-S (Zhu et al., 2024) | 80.6 | 69.4 | 86.0 | 20.3 | 45.8 | 33.4 |
| Vim-B (Zhu et al., 2024) | 81.2 | 70.0 | 86.2 | 27.5 | 46.0 | 33.9 |
| ARM-B | 83.2 | 72.3 | 88.0 | 31.9 | 48.9 | 37.2 |
| Vim-L (Zhu et al., 2024) | 81.0 | 69.8 | 86.0 | 27.9 | 44.7 | 31.8 |
| ARM-L | 84.5 | 74.0 | 88.6 | 41.4 | 52.1 | 39.2 |
| ARM-H | 85.0 | 75.6 | 89.2 | 42.3 | 53.2 | 40.5 |

Table 4: Object detection and instance segmentation use Mask R-CNN on COCO dataset at a resolution of 1024×1024. Semantic segmentation uses the UperNet framework on the ADE20K dataset at a resolution of 512×512.

| Method | Object detection $AP^{box}$ | Instance segmentation $AP^{mask}$ | Semantic segmentation mIoU |
|---|---|---|---|
| Vim-S (Zhu et al., 2024) | 43.2 | 40.0 | 44.9 |
| Vim-B | N/A | N/A | 45.2 |
| DeiT-B (Touvron et al., 2020) | 46.8 | 41.5 | 45.5 |
| MambaMLP | N/A | N/A | 45.3 |
| ARM | 49.2 | 43.9 | 47.7 |

Further enhancements are observed when ARM-B is finetuned with increased input sizes of 384×384 and 448×448 with the patchify stride of 8, where performance improves to 84.2% and 85.2%, respectively. We also report the comparison to VMamba-B, which takes a hybrid architecture: When configured with inputs of 224×224, ARM-B slightly underperforms VMamba-B by 0.7% but enjoys a much faster throughput, *i.e.*, ~4× faster; ARM-B with the inputs of 384×384 outperforms Vmamba-B by 0.3% and still maintains a faster throughput, *i.e.*, 440 imgs/s *vs.* 315 imgs/s.

Next, we scale the Mamba architectures to much larger model sizes. First, we observe that Mamba-based Vim sees a performance dip with the large size and fails to train stably at the huge size. This observation suggests that these prior Mamba-based architectures grapple with scaling challenges. Contrarily, ARM models excel in scalability — ARM-L achieves an accuracy of 84.5%, marking a 3.5% improvement over Vim-L, and ARM-H sets a new benchmark for the largest Mamba architecture in vision to date by reaching 85.0% accuracy. Moreover, by tuning ARM at a larger resolution of $384 \times 384$, further leveraging the model's capacity to handle long sequences at a linear complexity, we observe additional gains: a 0.6% increase with ARM-L and a 0.5% increase with ARM-H. Notably, ARM-H attains the best Mamba accuracy of 85.5% on ImageNet classification.

## 4.3 ROBUSTNESS

We report the "zero-shot" robustness evaluation (*i.e.*, no finetuning on these ImageNet variants) of Mamba architectures in Table 3. We can observe that ARM consistently shows much stronger robustness than the supervised Vim by, *e.g.*, ARM-B exhibits improvements ranging from 1.8% to 4.4% over supervised Vim-B across these robustness benchmarks. More impressively, ARM-L extends these gains even further, showing enhancements ranging between 2.6% and 7.4% when compared to supervised Vim-L. In addition, ARM-H, our largest model variant, not only continues this trend but also shows an average performance superiority of 1.1% over ARM-L, reaffirming the efficacy of scaling up the model size on enhancing robustness.

## 4.4 DOWNSTREAM GENERALIZATION

**Object Detection and Instance Segmentation** We benchmark object detection and instance segmentation performance on COCO 2017 (Lin et al., 2014) by integrating our ARM as backbone into Mask R-CNN Following Swin Transformer's protocol (Liu et al., 2021), we finetuning on COCO with 36 epochs, multi-scale training with shorter edges randomly resized between 480-800 pixels. Optimization employs AdamW (weight decay 0.05, batch size 16, initial learning rate 1e-4), with consistent augmentation strategies across all backbones to ensure comparability. As shown in Table 4, ARM achieves 49.2 $AP^{box}$ on object segmentation and outperforms DeiT-B and Vim-S by 2.4 and 6.0 $AP^{box}$. Similarly, on instance segmentation, ARM achieves 49.2 $AP^{mask}$ on object segementation and outperform DeiT-B and Vim-S by 2.4 and 6.0 $AP^{mask}$.

Table 5: Ablation on the number of predictions units.

| Num of Prediction unit | Cluster size | Top-1 (%) |
|:---:|:---:|:---:|
| 0 (Supervised) | N/A | 81.2 |
| 144 (iGPT) | $1 \times 1$ (Pixel) | 79.8 |
| 4 | 96×96 | 82.0 |
| 9 | 64×64 | 82.5 |
| 16 | 48×48 | 82.2 |
| 36 | 32×32 | 81.9 |
| 144 | 16×16 | 81.7 |

Table 6: Ablation on prediction orders.

| Order | Direction | Top-1 (%) |
|:---:|:---:|:---:|
| Row-first | Forward | 82.5 |
| Row-first | Backward | 82.3 |
| Column-first | Forward | 82.5 |
| Column-first | Backward | 82.4 |
| Random | Random | 81.5 |

**Semantic Segmentation** We hereby report ARM's performance on ADE20K semantic segmentation (Zhou et al., 2019). We adopt our ARM-B as the backbone and UperNet as the framework. The implementation details align with established practices from (Wang et al., 2021; Ren et al., 2022b; 2023c) and the mmsegmentation toolkit (Contributors, 2020). We employ the AdamW optimizer with a weight decay coefficient of 0.01 over 160,000 iterations. The learning rate is initialized at 6e-5, preceded by a 1,500-iteration warm-up phase, followed by a linear decay schedule. We incorporate standard data augmentation techniques like random horizontal flipping and multi-scale resizing. Training is conducted on $512 \times 512$ resolution crops, while inference is performed at both single-scale and multi-scale resolutions to evaluate generalization. As shown in Table 4, ARM achieves 47.7 mIoU on semantic segmentation and outperforms DeiT-B and Vim-S by 2.2 and 2.8. Under identical architecture, ARM outperform MambaMLP by 2.4 mIoU.

## 4.5 ABLATION STUDY

Unless otherwise specified, all ablations are performed on ARM-B under 300 epochs pretraining.

**Number of prediction units.** Table 5 reports the ablation on the number of prediction units. We start from the cluster size equal to the patch size (*i.e.*, each cluster contains only one patch), resulting in a total of 144 prediction units. We note that, even with this vanilla setup, autoregressive pretraining successfully helps MambaMLP improve performance from 81.2% (via supervised training) to 81.7%. Then, we gradually group multiple patches into one cluster, thereby reducing the total number of prediction units. We note that the performance first increases and then decreases — the best performance is achieved when the number of the prediction units is set to 9, corresponding to a cluster size of 64×64. Specifically, this setup provides a performance improvement of 1.3% over the supervised counterpart and 0.8% over the vanilla autoregressive pretrained counterpart (*i.e.*, with a cluster size of 144). We also report the comparison to MambaMLP trained under the iGPT-style autoregressive pretraining — with the input image size at $144 \times 144$ and setting per pixel as the prediction unit, it underperforms our best setup by 2.7% (*i.e.*, 79.8% *vs.* 82.5%).

**Prediction Order.** Table 6 shows that different pre-defined orders only lead to minor performance variances. For example, both row-first and column-first forward prediction orders achieve an identical performance of 82.5%; even the least favorable case, where the prediction order was row-first and backward, only underperforms the best case by 0.2%. Nonetheless, if we do not predefine the prediction order and pick a random permutation, the performance significantly drops to 81.5%.

**Decoder Design.** Our exploration into decoder design is summarized in Table 7. We first focus on the *decoder depth*, finding that increasing the depth up to 4 progressively enhanced performance up to 82.5%; further increasing the decoder depth to 8 sees a performance saturation. With this 4-layer decoder setup, we next study the width of the decoder. By ablating these three options {384, 512, 1024}, we empirically observe that setting the decoder width to 512 yields optimal accuracy.

**Prediction targets.** We hereby explore different prediction targets for our ARM. By default, we use per-patch normalized pixels with mean square error (MSE) loss. For comparison, we ablate it against two setups: 1) unnormed pixels with MSE loss, and 2) discretized tokens of the patches derived from dVAE (Bao et al., 2022) with cross-entropy loss. The results, presented in Table 8, show that employing normalized pixels as the target with MSE loss yields the best performance, achieving an accuracy of 82.5%. Comparatively, this configuration outperforms the model using discrete tokens from dVAE by 0.3% and the model leveraging unnormed pixels which trailed by 0.6%.

Table 7: Ablation on decoder designs.

| Dec. Depth | Dec. Width | Top-1 (%) |
|---|---|---|
| 1 | 512 | 82.1 |
| 2 | 512 | 82.4 |
| 4 | 512 | 82.5 |
| 8 | 512 | 82.5 |
| 4 | 384 | 82.3 |
| 4 | 512 | 82.5 |
| 4 | 1024 | 82.2 |

Table 8: Ablation on prediction targets.

| Targets | Top-1 (%) |
|---|---|
| dVAE (Bao et al., 2022) | 82.2 |
| Pixel (He et al., 2022) | 81.9 |
| Normed Pixel (He et al., 2022) | 82.5 |

Table 9: Comparison of architecture and pretraining paradigms. FPS represents the inference speed after supervised finetuning of the model. The † symbol indicates that Vim, when subjected to contrastive learning, experiences poor performance, potentially due to mode collapse.

| Architecture | Pretraining paradigm | Training Cost (h) ↓ | FPS (imgs/s) ↑ | Top-1 (%) |
|---|---|---|---|---|
| MambaMLP | Supervised | 110 | 1330 | 81.2 |
| MambaMLP | Contrastive | 330 | 1330 | 81.4 |
| MambaMLP | MAE | 70 | 1330 | 81.6 |
| MambaMLP | ARM | 34 | 1330 | 82.5 |
| Vim | Supervised | 165 | 923 | 81.2 |
| Vim | Contrastive | 510 | 923 | 80.2† |
| Vim | MAE | 106 | 923 | 81.4 |
| Vim | ARM | 57 | 923 | 82.2 |

**Pretraining paradigm.** As shown in Table 9, we evaluate different pretraining paradigms, including contrastive learning (Chen et al., 2021), MAE (He et al., 2022), and our ARM. Firstly, we note that all pretraining methods result in performance gains over the supervised counterpart, demonstrating the benefits of self-supervised visual pretraining on Mamba architectures. However, using MAE or contrastive learning, the performance is only moderately improved by 0.4% and 0.2%, respectively, over the supervised baseline. In contrast, our ARM achieves significant improvements of 1.3% over the supervised baseline, as well as achieves higher accuracy than both contrastive learning and MAE. Additionally, in terms of efficiency, ARM requires just 34 hours of pretraining, cutting the training duration in half compared to MAE, which is already noted for its relatively low pretraining demands.

**Architecture design.** Exploring further into architectural impacts, Table 9 (from the 5th row to the 8th row) presents our investigation into whether Vim, another variant within the Mamba architecture, benefits from autoregressive pretraining. Results indicate a positive response as ARM-trained Vim reaches an 82.2% accuracy on ImageNet, marking a 1.0% improvement over its supervised-only counterpart. Contrastingly, other pretraining paradigms did not fare as well for Vim: when subjected to contrastive learning, Vim experiences training instability, falling below the supervised baseline; MAE pretraining on Vim only slightly improved over the supervised method, with a marginal gain of 0.2%. These results further support the effectiveness of ARM in pretraining Mamba in Vision.

As a side note, it is important to highlight that although Vim's performance improves with ARM pretraining, it operates ~45% slower during inference compared to MambaMLP. Additionally, MambaMLP incurs only ~66% of the training cost required for pretraining Vim under the ARM framework. These points underscore the superior efficiency of our default ARM framework.

## 5 CONCLUSION

This study introduces a novel autoregressive visual pretraining strategy tailored for Mamba architectures. Through this method, we have significantly improved the scalability and benchmark performance of Mamba-based models, setting new standards in their operational functionality. We hope this work can lay a strong foundation for future explorations and potential expansions in the usage of autoregressive pretraining strategies for Mamba architectures within the vision community.

**Acknowledge** This work is supported by ONR with N00014-23-1-2641. We thank the AWS Cloud Credit for Research program for partially supporting the computation of this project.

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

# A  APPENDIX

## A.1  COMPARED WITH OTHER AUTOREGRESSIVE MODELING FOR SELF-SUPERVISED LEARNING

We compare our ARM with AIM (El-Nouby et al., 2024) which naively flatten patches in a raster order (row by row), ours introduces a more strategic approach by grouping spatially neighboring patches into larger clusters and using these clusters as prediction units. As shown both in our ablation and recent work (Hu et al., 2024), selecting the appropriate sequence order is crucial for achieving high performance in autoregressive models — e.g., all these works find a better scan strategy than the raster scan (used in AIM) in achieving higher performance. Moreover, we finetune AIM-0.6B on ImageNet-1K. Specifically, we used the AdamW optimizer for 50 epochs, a peak learning rate of 2e-3 with cosine decay, a warm-up period of 5 epochs, and a batch size of 1024. Data augmentation and regularization techniques included a drop path rate of 0.3, RandAugment, label smoothing of 0.1, Mixup at 0.8, and CutMix at 1.0.

We can observe that ARM reaches 85. 0% Top-1 accuracy and makes a 0.8% improvement over AIM-0.6B and a 1.2% improvement over AIM-0.6B when causal masking is used.

Table 10: Comparison with AIM.

| Model | Top-1 Accuracy (%) |
|---|---|
| AIM-0.6B (causal) | 83.8 |
| AIM-0.6B | 84.2 |
| ARM-H (ours) | 85.0 |

## A.2  STATISTICAL SIGNIFICANCE

The default seed used in our experiments is 0 as previous work (He et al., 2022; Zhu et al., 2024; Liu et al., 2024b). We run 3 additional experiments with seeds = 1024, 9718, 87144, and report the results in Table 11. Cluster deign is stable and make consistent improvements by 0.8%.

Table 11: Statistical significance with different seeds.

| Prediction Unit | Top-1 Accuracy (%) |
|---|---|
| cluster | $82.5 \pm 0.1$ |
| token | $81.7 \pm 0.2$ |

