# OpenReview forum: "Autoregressive Pretraining with Mamba in Vision"
_ICLR.cc/2025/Conference — ICLR 2025 Poster_

### Official Review · Reviewer_Xz8m · 2024-10-23

**Soundness:** 2
**Presentation:** 2
**Contribution:** 2
**Rating:** 6
**Confidence:** 4

**Summary:**

The paper introduces autoregressive pretraining to the domain of vision mamba models. Drawing inspiration from "Scalable Pre-training of Large Autoregressive Image Models" (AIM), it shows how the method naturally adapts to Mamba models, which are inherently autoregressive, at least in their simplest form. This pretraining method shows positive scaling traits, improving downstream classification accuracy, and also allowing for larger variants to be trained than previously possible.

**Strengths:**

The paper finds an effective way to apply AIM to the space of vision-mamba variants, demonstrating that it improves the pareto frontier for mamba-based vision models. The pretraining unlocking the use of larger vision-mamba variants is a useful outcome.

**Weaknesses:**

At a high level, the novelty in this paper is quite limited. It appears to mostly be “AIM [1], but with Mamba”. Similarly, I think table 2 needs to include AIM; I know this is difficult because they don’t provide finetuned accuracy numbers (only probes), but they do provide the model checkpoints, so even a casual attempt at finetuning their 0.6B model on imagenet would be helpful to contextualize on the huge models.

Given that ARM isn’t bringing anything new to the theory of SSSMs or autoregressive pretraining, it doesn’t feel like equations 1-6 are necessary, or at least could be greatly condensed [1, 2], . Particularly given the degree to which this paper builds on AIM, leveraging their section 4.1 would be better. Similarly, equations 7-9 are highly redundant. Distilling all of this down, all of the equations presented in this paper don’t seem to aid in clarity.

“Prediction units”. This is defined in 3.2.2 as generally being patches, aside from iGPT which uses pixels. This seems to be reinforced by figure 2d and figure 3. However, section 4.4.1 confusingly switches to clusters being prediction units. Based on the figures, it seems like a patch is always a prediction unit. But a good amount of the paper is spent on processing order, with a distinction between tokens and clusters. Equation 9 seems to suggest that a full cluster is predicted simultaneously. Presumably ARM is patch based though, so it seems like clusters are mostly just a way to define a particular order of patches, and due to scan being an enforced sequential algorithm, a cluster isn’t actually predicted simultaneously. This would make figure 3 make sense.

Figure 4: I’m struggling to find anywhere where the switch from 1 scan to 4 scans is justified. It would be good to at least demonstrate the difference in accuracy between 1 scan and 4 in your setting.

Table 4 has clarity issues. It would be helpful to directly specify the cluster size instead of having it be $\frac{cluster\\_size}{patch\\_size}$.

[1] Scalable Pre-training of Large Autoregressive Image Models (http://arxiv.org/abs/2401.08541)

[2] Mamba: Linear-Time Sequence Modeling with Selective State Spaces (https://arxiv.org/abs/2312.00752)

**Questions:**

4.4.4 Which normalization method are you referring to for pixel targets? I’m guessing you’re referring to MAE’s [3] method, but this needs to be explicitly cited.

Do you have any experiments where you just vary the random seed to get a sense of the variance in model quality just due to randomness? For example, there are 6 examples of the word “significant”, but I assume this is not statistical significance. Taking table 4 as an example, there is a spread of 0.8% between a 1x1 cluster and a 4x4 cluster. Knowing the variance of your method on Top-1% would be really helpful for determining whether these difference are actually significant.

[3] Masked Autoencoders Are Scalable Vision Learners (http://arxiv.org/abs/2111.06377)

---

> ### Author Response · Authors · 2024-11-23
> **Rebuttal by authors**
>
> Q1: Compare with AIM
>
> A1: Thanks for raising this concern. We would like to clarify that, while these two works are highly related, our ARM is significantly different from AIM. While AIM naively flatten patches in a raster order (row by row), ours introduces a more strategic approach by grouping spatially neighboring patches into larger clusters and using these clusters as prediction units. As shown both in our ablation and recent work like [1,2], selecting the appropriate sequence order is crucial for achieving high performance in autoregressive models — e.g., all these works find a better scan strategy than the raster scan (used in AIM) in achieving higher performance.
> Moreover, we follow your suggestion to finetune AIM-0.6B on ImageNet-1K. Specifically, we used the AdamW optimizer for 50 epochs, a peak learning rate of 2e-3 with cosine decay, a warm-up period of 5 epochs, and a batch size of 1024. Data augmentation and regularization techniques included a drop path rate of 0.3, RandAugment, label smoothing of 0.1, Mixup at 0.8, and CutMix at 1.0. We report the experiment results below:
>
> | Model | Top-1 Accuracy (%) |
> |---------------------|--------------------|
> | AIM-0.6B (causal) | 83.8 |
> | AIM-0.6B | 84.2 |
> | **ARM-H (ours)** | **85.0** |
>
>
> We can observe that ARM achieves a 0.8% improvement over AIM-0.6B and a 1.2% improvement over AIM-0.6B when causal masking is used.
>
> We will add these discussions and results in the next version to clearly highlight the differences between our work and AIM.
>
>
> Q2: all of the equations presented in this paper don’t seem to aid in clarity.
>
> A2: Thanks for your suggestion. As Mamba is still a relatively new architecture, we believe including Equations 1 to 4 can provide useful background for readers to understand them. However, we are open to moving them into the supplementary material given the slight redundancy.
>
> Regarding Equations 5 - 9, we believe these are important for illustrating our step-by-step adaptation of autoregressive modeling to our proposed ARM method. But we are also open to revisiting these equations to see if we can re-deliver them in a more concise format.
>
>
> Q3: Confusion of “Prediction units”.
>
> A3: Thanks for bringing up this confusion. Yes, your understanding is correct — the image patch is our basic prediction unit, and clusters are mostly just a way to define a particular scan order of patches. We will revise the corresponding equations, figures, and text descriptions to more consistently and clearly deliver this point.
>
> Q4: Switch from 1 scan to 4 scan.
>
> A4: Sorry for the confusion. Firstly, we would like to clarify that using one scan is a hard requirement in pretraining. This is because autoregressive modeling requires a strict causal structure, where each token is predicted based only on past tokens in the defined sequence order. If multiple scans are used, this autoregressive nature will break and information leakage will happen, resulting in the learned feature representation collapsing. In the finetuning stage, four scans are used mainly to enhance performance, following the setup in Vim. In contrast, if only one scan is used, the performance will drop by 0.4%, from 82.5% to 82.1%.
>
> Q5: It would be helpful to directly specify the cluster size instead of having it be $\frac{cluster\\_ size}{pathch\\_size}$.
>
> A5: Yes, in Table 4, our second column directly reports the setup of cluster size.
>
> Q6: normalization method
>
> A6: Thanks for the suggestion. Yes, your understanding is correct and we will add the reference for better clarity.
>
> Q7: Do you have any experiments where you just vary the random seed to get a sense of the variance in model quality?
>
> A7: Thanks for your suggestion. We first clarify that the default seed used in our experiments is 0. Following your suggestion, we run 3 **additional** experiments with seeds = {1024, 9718, 87144}, and report the results below
>
> These experiments demonstrate that our results are stable and that the performance improvements are not due to random chance. We will include these results and a discussion on statistical significance in the next version
> | Prediction Unit | Top-1 Accuracy (%) |
> |---------------------|--------------------|
> | cluster|82.5±0.1 |
> |token |81.7 ±0.2 |
> These experiments demonstrate that our results are stable and that the performance improvements are not due to random chance. We will include these results and a discussion on statistical significance in the next version
>
> [1] Hu, V. T., Baumann, S. A., Gui, M., Grebenkova, O., Ma, P., Fischer, J., & Ommer, B. (2025). Zigma: A dit-style zigzag mamba diffusion model. In European Conference on Computer Vision (pp. 148-166). Springer, Cham.
>
> [2] Yu, Q., He, J., Deng, X., Shen, X., & Chen, L. C. (2024). Randomized Autoregressive Visual Generation. arXiv preprint arXiv:2411.00776.
>
> [3] VMamba: Visual State Space Model

---

> > ### Comment · Reviewer_Xz8m · 2024-11-25
> >
> > Thank you for the clarifications. I feel as though the work has merit, and want to encourage that, however, I also strongly encourage the authors to take the review feedback on draft clarity to heart, and incorporate the rebuttal data into the manuscript.

---

> > > ### Author Response · Authors · 2024-11-27
> > > **Thanks!**
> > >
> > > Thanks for supporintg our submision, and raising your score from 3 to 6.
> > >
> > > Yes, we can assure you that we are carefully taking all these comments, and will certainly incorporate them into the final version to strengthen our paper.
> > >
> > > We thank again for your helpful review.

---

### Official Review · Reviewer_mXuW · 2024-10-30

**Soundness:** 3
**Presentation:** 3
**Contribution:** 3
**Rating:** 6
**Confidence:** 4

**Summary:**

This paper applies autoregressive pretraining to Mamba for visual classification tasks.  The Mamba-based architecture uses a block similar to Visual Mamba (VIM) and VMamba, MambaMLP, that applies Mamba over a patch sequence ordered by a 2-level "clustered" raster order and MLP over channels (though, I may have misunderstood the channel vs sequence decomposition, see questions below).  Multiple ablations are performed on Imagenet1k, profiling the system's behavior and performance; substantial improvements are found from using autoregressive pretraining on Imagenet1K and several of its variants.

**Strengths:**

The experiments clearly show the benefit of autoregressive pretraining for this task, on both MambaMLP (the proposed architecture here) and Vim.  Ablations on token cluster size (a nice idea), decoder and patch representations, and sequence order describe the system behavior well.  Overall, the presentation is quite clear and the system well described, though there are key points that I think could be explained better (see weaknesses below).

**Weaknesses:**

There are several important pieces of the method that lacked detailed explanation:

* The MambaMLP block is described only in a sketch diagram (Fig 4) and a sentence in sec 3.3 ("uses Mamba as the token mixer and MLP as the channel mixer").  This is a good summary, but didn't explain the block architecture well enough for me to understand the details:  Does this mean Mamba is applied independently to each channel and alternates with MLP?  And how are the Delta, B and C set (which inputs and which channels)?  A more explicit explanation, making use of the summary of Mamba in Sec 3.1 and showing where and how this is used in MambaMLP in addition to MLP would be useful here.

* Likewise, there is no explanation of the patch decoder --- there is an ablation on its design, but I didn't find anything saying what architecture was used.

* Another key point, is how can only one scan order be used in pretraining but four in fine-tuning?  If the other 3 scan orders aren't pretrained, how are they initialized?  And what is the impact of using only one scan order (the one that was pretrained) in fine-tuning (and conversely, is it possible somehow to pretrain all four orders)?  (The terms "scan" and "expand" are assumed and not summarized or used in an explanatory context, either)

A second, more minor weakness is that only image classification is considered, and no other tasks.  Additional task applications would strengthen the results, though the existing ones with classification are already convincing to show the merits of the method.

**Questions:**

See above.  Additional, more minor questions/comments below.

* Additional related work on autoregression in vision might be mentioned and compared, e.g.  Bai et al 2023 "Sequential Modeling Enables Scalable Learning for Large Vision Models"; Yu et al 2021 "Diverse image inpainting with bidirectional and autoregressive transformers"; Esser et al 2021 "Taming transformers for high-resolution image synthesis."

* Could use better background summary for what is meant by "scan" and "expand" in Fig 4 / Sec 3.3.  While the basic SSM is summarized, as are the orderings in sec 3.2, a little more explanation of the "scans" here would be useful

* Sec 4.3:  When applied to other datasets, is autoregressive pretraining performed only on imagenet1k, the target dataset, or both?

* Eq (3) says $h'(t)$ but this is the descretized system, so should this be $h_t$ on the first line ?

* Random order l.294:  To confirm, the order is a single random permutation order that is fixed for each model instance (so every time the model is called on an image, the same order is always used)?

* typo on l.470, should say width is 512 no depth

---

> ### Author Response · Authors · 2024-11-23
> **Rebuttal by authors**
>
> Q1: The MambaMLP block is described only in a sketch diagram (Fig 4) and a sentence in sec 3.3 ("uses Mamba as the token mixer and MLP as the channel mixer"). This is a good summary. Does this mean Mamba is applied independently to each channel and alternates with MLP?
>
> A1: Sorry for the confusion. We would like to clarify that, in our architecture, we employ Mamba as the token mixer, which mixes information among different tokens to capture their relationships. Or in simpler terms, by replacing self-attention with Mamba in the ViT, we obtain our MambaMLP. We will make this point clearer in the next version.
>
> Q2: The explanation of the patch decoder.
>
> A2: Sorry for missing this detail. We clarify that our decoder follows exactly the same design as in MAE [2], which uses Transformer blocks (by default, we choose depth=4 and width=512); Also in Table 6, we ablate different choices of decoder depth and width. Please note this decoder is dropped when finetuning to different downstream tasks.
>
> Q3: how can only one scan order be used in pretraining but four in fine-tuning? If the other 3 scan orders aren't pretrained, how are they initialized?
>
> A3: Sorry for the confusion. Firstly, we would like to clarify that using one scan is a hard requirement in pretraining. This is because autoregressive modeling requires a strict causal structure, where each token is predicted based only on past tokens in the defined sequence order. If multiple scans are used, this autoregressive nature will break and information leakage will happen, resulting in the learned feature representation collapsing. In the finetuning stage, four scans are used mainly to enhance performance, following the setup in Vim. In contrast, if only one scan is used, the performance will drop by 0.4%, from 82.5% to 82.1%.
>
> Regarding initialization, we clarify that the parameters of all four scans in finetuning are all initiated from the one used in pretraining.
>
> Q4: Additional task applications would strengthen the results.
>
> A4: Thank you for your suggestion! We have fine-tuned ARM-B on the semantic segmentation task and achieved a mIoU of 47.3%, which is a notable improvement over Vim-B (45.2% mIoU) and MambaMLP-B (45.3% mIoU). We are also experimenting with object detection setups, which require more training resources, and will provide those results as soon as they are available. If we cannot include them before the discussion deadline, we will ensure they are incorporated in the final revision.
>
>
> Q5: more related work.
>
> A5: Thanks for bringing in these related works which focus on leveraging autoregressive modeling for image generation. We will discuss them in the related work section of the revision.
>
> Q6:the scan and expand
>
> A6: Sorry for the confusion. The scan refers to the order of the sequence that the Mamba processes; and the expand refers to expanding the hidden states with more channels. We will clarify these concepts in the revision.
>
> Q7:Sec 4.3: When applied to other datasets, is autoregressive pretraining performed only on imagenet1k, the target dataset, or both?
>
> A7: No, our models are only pretrained on ImageNet-1k, and the accuracy on Tab 3 reflects the models’ zero-shot ability on these datasets. We will make it clearer in the next version.
>
> Q8: ht for the discretized system.
>
> A8: Yes, you are correct. We will fix this typo in the revision.
>
> Q9: Random order
>
> A9: This randomness is running on the fly, i.e., the same image will be processed in different orders at different training steps. We will make it clear in the next version.
>
> Q10: typo
>
> A9: Thanks! We will fix it in the next version.
>
>
> [1] Yu, W., Luo, M., Zhou, P., Si, C., Zhou, Y., Wang, X., ... & Yan, S. (2022). Metaformer is actually what you need for vision. In Proceedings of the IEEE/CVF conference on computer vision and pattern recognition (pp. 10819-10829).
> [2] He, K., Chen, X., Xie, S., Li, Y., Dollár, P., & Girshick, R. (2022). Masked autoencoders are scalable vision learners. In Proceedings of the IEEE/CVF conference on computer vision and pattern recognition (pp. 16000-16009).

---

> > ### Author Response · Authors · 2024-11-27
> >
> > Dear Reviewer mXuW,
> >
> > We sincerely appreciate your review. We have carefully considered each of your questions and provide detailed responses in the rebuttal. Please let us know if you have any further questions or concerns.

---

> > > ### Comment · Reviewer_mXuW · 2024-12-02
> > > **response**
> > >
> > > Thanks for the clarifications, the unclear pieces that I mentioned before are clearer now.  In particular that pretraining is applied only on Imagenet1k, not the target fine-tuning dataset, is good to see, as well as the initial results for segmentation tasks.  I will keep my 6 accept score --- also as other reviewers have mentioned, please be sure to include the revisions in these discussions into the paper.

---

> > > > ### Author Response · Authors · 2024-12-02
> > > > **Thanks!**
> > > >
> > > > Thanks for supporting our work. We are glad to see that all of your concerns have been addressed now.
> > > >
> > > > Yes, we can assure you that these additional experiments, discussions, and clarifications will be integrated into the final version to strengthen our paper.

---

### Official Review · Reviewer_BJtj · 2024-11-01

**Soundness:** 3
**Presentation:** 3
**Contribution:** 2
**Rating:** 5
**Confidence:** 3

**Summary:**

This paper investigates the Mamba architecture for visual data using autoregressive pre-training. It shows the Mamba’s visual capability can be enhanced through autoregressive pretraining. It finds autoregressive pretraining equips the Mamba architecture can obtain better performance compared to its supervised-trained counterparts, and also shows the potential for scaling, based on the experiments on ImageNet-1K.

**Strengths:**

This paper is overall writing well. This work to investigates the mamba architecture using autoregressive pre-training in visual data is important for the community.
The results are inspiring for the community, e.g., ARM can scale somewhat well, compared to the Vim counterparts; and the ablation on the prediction unit shows the effectiveness of the cluster-based prediction.

**Weaknesses:**

1.My main concern is the experimental setting:
(1)	When the results of the proposed method in Table 2 is good, it is not clear for the setting up of other supervised baselines (RegNetY-16G, DeiT-B, Vim-B), e.g, how many epochs these model training? Is it fair to compare these baselines without the same training epochs?
(2)	Besides, the protocol for evaluating self-supervised method is usually pre-trained on large-scale dataset, then evaluated on the downstream tasks (e.g., linear probe or finetuned with few-shot examples). It is weird that this paper pre-trained on ImageNet using self-supervised learning and fine-tuned with all the ImageNet examples. For a stronger result, I think this paper should pretrain on larger unlabeled dataset, then finetuned on ImageNet. In current experiments, the results are not strong, since Vits using self-supervised learning has shows its potential in performance and scaling ability. Another point, I think this paper should compare the state-of-the-art Vits with the same parameters in Table.2 (the parameters matters for performance), and I donot find the proposed method have better performance than Vits.


2.It is not clear why this paper uses different setting for pre-training (Fig.4 b) and fine-tuning (Fig.4 c) for the MambaMLP architecture. Is it from the empirical observation? If yes, it is better to show the results, when unifying the setting (e.g., using the same setting for pre-training and finetuning). It is weird that MambaMLP needs two configurations for pre-training and fine-tuning, especially in the situation that this paper only conducts experiments on one dataset (not generable but only tailored for one dataset)

**Questions:**

See weaknesses

---

> ### Author Response · Authors · 2024-11-23
> **Rebuttal by authors -- Part I**
>
> Q1: it is not clear for the setting up of other supervised baselines. Is it fair to compare these baselines without the same training epochs?
>
> A1: Sorry for the confusion. We can add one more column in Table 2 to report the training epochs. But we would like to point out that comparing training epochs alone may not provide a fair basis for evaluating different methods, especially when comparing self-supervised learning to supervised learning approaches. This is because, in self-supervised learning strategies like MAE, the inputs are typically masked at a very high ratio, resulting in more efficient training per epoch compared to supervised methods where the full input is processed.
>
> To provide a more equitable comparison, we believe that reporting the actual training cost in terms of computational resources (e.g., A5000 GPU hours) offers a more direct measure. The updated Table 2 with the training cost for the base-size Mamba is shown below:
>
> | Method | Pre-training Epochs | Fine-tuning Epochs | Training Cost (A5000 GPU hours) | Top-1 Accuracy (%) |
> |----------|---------------------|--------------------|---------------------------------|--------------------|
> | **ARM-B** | 1600 | 100 | 214 | **83.2** |
> | **ARM-B** | 300 | 100 | 71 | **82.5** |
> | Vim-B | N/A | 300 | 165 | 81.2 |
> | MambaMLP | N/A | 300 | 110 | 81.2 |
>
> From the updated table, we observe that in the original setup (i.e., 1600 epoch pretraining + 100 epoch finetuning, which follows MAE’s setup), ARM-B achieves a higher accuracy (83.2%) but with a higher training cost (214 hours on 8XA5000). To better demonstrate the efficiency of our method, we conducted an additional experiment with a shorter pre-training schedule of 300 epochs (totaling 300 pre-training + 100 fine-tuning epochs). This setup reduces the total training cost to 71 hours on 8XA5000, which is less than that of the supervised baselines, while still achieving a superior accuracy of 82.5%. This indicates that ARM-B can attain better performance even with reduced computational resources compared to the supervised methods.
> We will include these details and the updated Table 2 in the next version to enhance the clarity and fairness of our comparisons.
>
> Q2: The protocol for evaluating self-supervised method is usually pre-trained on large-scale dataset, then evaluated on the downstream tasks e.g., linear probe. I think this paper should pretrain on larger unlabeled dataset, then finetuned on ImageNet.
>
> A2: We appreciate your concern regarding the evaluation protocol. While early self-supervised learning methods like SimCLR [4] and MoCo [5] focused on linear evaluation or few-shot learning on downstream tasks, the evaluation protocols in recent research have evolved. Specifically, starting from works like BEiT [1], it has become a standard practice to pre-train models on ImageNet in a self-supervised manner and then fine-tune on the same ImageNet dataset using all available labels. This approach allows for a direct and fair comparison with supervised baselines under the same data constraints. Additionally, as shown in many recent works like [7] – the strong performance on ImageNet often positively correlates with generalization abilities when scaled to larger datasets.
>
> Therefore, we believe that our evaluation protocol is consistent with current research practices and sufficient to demonstrate the effectiveness of our method. We will add these clarifications in the next version to avoid future confusion.

---

> > ### Author Response · Authors · 2024-11-23
> > **Rebuttal by authors -- Part Ⅱ**
> >
> > Q3: I think this paper should compare the state-of-the-art Vits with the same parameters in Table.2 (the parameters matters for performance), and I donot find the proposed method have better performance than Vits.
> >
> > A3:  Thanks for raising this concern. While we acknowledge the importance of achieving sota performance, we stress that the primary objective of our work is to develop an effective pre-training strategy for the Mamba architecture, which we believe is a promising alternative to Transformers. Specifically, instead of using contrastive learning or masked image modeling widely adopted in ViTs as the pretraining strategy, we choose to pretrain Mamba using autoregressive modeling — this is because Mamba architectures are inherently well-suited for autoregressive modeling, which methodically constructs token-wise relationships in a strictly progressive and unidirectional manner.
> >
> > While our method does not yet surpass state-of-the-art ViTs in terms of absolute performance, we believe that exploring and improving Mamba is valuable for the community. Mamba models offer unique advantages, particularly in terms of computational efficiency. The self-attention mechanism in Transformers has quadratic computational complexity with respect to sequence length, which poses challenges for processing long sequences or high-resolution inputs. In contrast, the Mamba architecture has linear computational complexity, enabling more efficient handling of high-resolution images and finer-grained representations (e.g., 448×448 inputs with an 8×8 patch size in our experiment). We believe these computational benefits make the Mamba architecture a promising candidate for tasks involving high-resolution images, 3D data, and videos.
> > We will make these points clearer in the next version.
> >
> > Q4: different settings for pre-training (Fig.4 b) and fine-tuning (Fig.4 c) for the MambaMLP
> >
> > A4: Sorry for the confusion. During the pre-training phase, we use only one scan in the MambaMLP architecture. This design choice is driven by the nature of autoregressive modeling, which requires a strict causal structure, i.e., each token is predicted based only on past tokens in the defined sequence order. If multiple scans are used, this autoregressive nature will break and information leakage will happen, resulting in the learned feature representation collapsing. Additionally, using one scan is more computationally efficient, which is advantageous during the extensive pre-training stage.
> >
> > In the fine-tuning phase, our goal shifts to achieving optimal performance on the downstream task. To capture richer and more holistic representations of the input data, we utilize four scans, as is standard in Vim [6]. Multiple scans allow the model to process information from all parts of the image bidirectionally, thereby enhancing its ability to understand complex patterns and relationships within the data (e.g., if only scan is used in finetuning, the performance will drop by 0.4%, from 82.5% to 82.1%).
> >
> > We will make these points clear in the next version to better clarify the reasoning behind the different configurations of MambaMLP.
> >
> > [1] Bao, H., Dong, L., Piao, S., & Wei, F. BEiT: BERT Pre-Training of Image Transformers. In International Conference on Learning Representations.
> >
> > [2] He, K., Chen, X., Xie, S., Li, Y., Dollár, P., & Girshick, R. (2022). Masked autoencoders are scalable vision learners. In Proceedings of the IEEE/CVF conference on computer vision and pattern recognition (pp. 16000-16009).
> >
> > [3] Dong, X., Bao, J., Zhang, T., Chen, D., Zhang, W., Yuan, L., ... & Guo, B. (2023, June). Peco: Perceptual codebook for bert pre-training of vision transformers. In Proceedings of the AAAI Conference on Artificial Intelligence (Vol. 37, No. 1, pp. 552-560).
> >
> > [4] Chen, T., Kornblith, S., Norouzi, M., & Hinton, G. (2020, November). A simple framework for contrastive learning of visual representations. In International conference on machine learning (pp. 1597-1607). PMLR.
> >
> > [5] He, K., Fan, H., Wu, Y., Xie, S., & Girshick, R. (2020). Momentum contrast for unsupervised visual representation learning. In Proceedings of the IEEE/CVF conference on computer vision and pattern recognition (pp. 9729-9738).
> >
> > [6] VMamba: Visual State Space Model
> >
> > [7] Singh, M., Duval, Q., Alwala, K. V., Fan, H., Aggarwal, V., Adcock, A., ... & Misra, I. (2023). The effectiveness of MAE pre-pretraining for billion-scale pretraining. In Proceedings of the IEEE/CVF International Conference on Computer Vision (pp. 5484-5494).

---

> > > ### Author Response · Authors · 2024-11-27
> > >
> > > Dear Reviewer BJtj,
> > >
> > > We sincerely appreciate your review. We have carefully considered each of your questions and provide detailed responses in the rebuttal. Please let us know if you have any further questions or concerns.

---

> > > ### Comment · Reviewer_BJtj · 2024-11-28
> > > **Comments for the response**
> > >
> > > I thank the responses of the authors. My minor concerns on the different setting for pre-training and fine-tuning is addressed. However, I main concerns on the protocol for evaluating self-supervised method still hold. The response "the strong performance on ImageNet often positively correlates with generalization abilities when scaled to larger datasets." is not closely related to my concern. We know self-supervised learning aims to learn good representation, which is for all kinds of task. If one pre-trains on Dataset A  then finetunes n Dataset A, it is not strong to support the goals of self-supervised learning. Currently, I keep my score.

---

> > > > ### Author Response · Authors · 2024-11-28
> > > > **evaluation on generalization**
> > > >
> > > > Dear Reviewer BJtj
> > > >
> > > > Thanks for your feedback and for giving us the opportunity to clarify our work further. We apologize for misunderstanding your prior concern regarding the generalization evaluation for our self-supervised method. To address it, we would like to highlight two key experiments in our work that demonstrate the effectiveness of our learned representations beyond the pre-training dataset.
> > > >
> > > > **1. Zero-Shot Classification on ImageNet Variants:** In the main paper (Table 3), we evaluate the zero-shot classification performance of our model on several challenging ImageNet variants, including ImageNet-V2, ImageNet-Real, ImageNet-A, ImageNet-R, and ImageNet-Sketch. These datasets introduce distribution shifts and adversarial examples that differ significantly from the original ImageNet data. The models are **pre-trained on ImageNet** and then **directly tested on these variants without any fine-tuning**, which is a popular and standard protocol for assessing generalization and robustness in representation learning (e.g., in [1]). We can observe that our ARM model demonstrates substantially stronger performance on these datasets compared to baseline models.
> > > >
> > > > **2. Transfer Learning to ADE20K Semantic Segmentation:** We have also conducted transfer learning experiments where we **fine-tune the pre-trained models on the ADE20K semantic segmentation task** (as detailed in our rebuttal, Response A4 to Reviewers mXuW and xhQF). ADE20K is a widely-used dataset for semantic segmentation and differs both in data domain and task objective from ImageNet classification. As shown in prior works [1,2], this transfer learning evaluation is also standard practice for measuring the generalization capability of self-supervised learning methods. Specifically, our finetuned ARM-B achieved a mIoU of 47.3%, which is a notable improvement over Vim-B (45.2% mIoU) and MambaMLP-B (45.3% mIoU).
> > > >
> > > > We will carefully re-clarify/integrate these experiments, which can strongly support the generalization of our ARM, in the revision. We hope this addresses your concern.
> > > >
> > > >
> > > > [1] He, K., Chen, X., Xie, S., Li, Y., Dollár, P., & Girshick, R. (2022). Masked autoencoders are scalable vision learners. In Proceedings of the IEEE/CVF conference on computer vision and pattern recognition (pp. 16000-16009).
> > > >
> > > > [2] Bao, H., Dong, L., Piao, S., & Wei, F. BEiT: BERT Pre-Training of Image Transformers. In International Conference on Learning Representations.

---

> > > > > ### Author Response · Authors · 2024-11-30
> > > > >
> > > > > Dear Reviewer BJtj,
> > > > >
> > > > > Thanks again for your comments. Could you please let us know if these additional experiments address your concerns on the evaluation of generalization.
> > > > >
> > > > > Thanks

---

> > > > > > ### Author Response · Authors · 2024-12-02
> > > > > > **Follow up on the generalization concern**
> > > > > >
> > > > > > Dear Reviewer BJtj,
> > > > > >
> > > > > > As the discussion deadline is approaching, we would like to follow up to see whether our response could address/mitigate your concerns about generalization.
> > > > > >
> > > > > > Looking forward to your response, and we are happy to provide more info if needed.
> > > > > > Thanks, Authors of 7773

---

### Official Review · Reviewer_xhQF · 2024-11-03

**Soundness:** 3
**Presentation:** 3
**Contribution:** 2
**Rating:** 6
**Confidence:** 4

**Summary:**

This paper shows that Vision Mamba can be enhanced through autoregressive pre-training, similar to using MAE pre-training to improve ViT. It uncovers two key recipes: (a) grouping spatially neighboring patches, and (b) vanilla ordering (row-by-row scan) is effective enough. It achieves significant improvement over supervised baselines.

**Strengths:**

- The paper is well written, including all technical details. It should be straightforward to implement.
- It is good to see Mamba follow similar pre-training conclusion with other vision architecture (CNN, Transformer): masked prediction in pre-training followed by fine-tuning boosts performance.
- It is good to know that cluster size plays an important role in pre-training.

**Weaknesses:**

- Figure 3 is a bit confusing. It seems to illustrate a case of 4 clusters, each has 4 patches. I assume that the arrows within a cluster show the order to flat patches, while the arrows across clusters show the order of prediction. If so, please use different colors and add explanation in the caption.
- In Table 2, please distinguish methods between supervised and pre-training+supervised fine-tuning. In addition, what is the performance for MambaMLP in huge size?
- Section 4.4, is it possible to improve random-order prediction by incorporating position encoding?

**Questions:**

- As shown in MAE, pre-training has another benefit to improve downstream tasks (like object detection, segmentation). Does this conclusion hold for ARM?
- It would be better to discuss a comparison with a prior work, “Image as First-Order Norm+Linear Autoregression: Unveiling Mathematical Invariance” (https://arxiv.org/pdf/2305.16319), which uses first-order autoregression to pre-train CNN architectures.

---

> ### Author Response · Authors · 2024-11-23
> **Rebuttal by Authors**
>
> We first thank the reviewer for the detailed comments and the appreciation of our work. We address the concerns below:
>
> Q1: Figure 3 is a bit confusing.
>
> A1: Thank you for pointing this out. You are correct—the arrows within each cluster indicate the order in which patches are flattened, and the arrows across clusters represent the order of cluster-wise prediction. We will follow your suggestion to use different colors for arrows and add corresponding explanations in the captions in the next version to improve the overall clarity of Figure 3.
>
> Q2: In Table 2, please distinguish methods between supervised and pre-training+supervised fine-tuning. In addition, what is the performance for MambaMLP in huge size?
>
> A2: Thanks for the suggestion! we will add one more column to mark whether the method is supervised or pre-training+supervised fine-tuning. Regarding the performance of MambaMLP at the huge size, it achieves an accuracy of 82.0%, which is 3.0% lower than our ARM-H. We will include this result in the next version.
>
> Q3: Section 4.4, is it possible to improve random-order prediction by incorporating position encoding?
>
> A3: Thanks for the suggestion! Actually, incorporating position encoding before the random permutation is our default setting for the random permutation scenario. Despite this, as presented in the paper, the random permutation still performs worse than our proposed row-first order. We will clarify this point more explicitly in the next version to avoid any confusion.
>
>
> Q4: pre-training has another benefit to improve downstream tasks (like object detection, segmentation). Does this conclusion hold for ARM?
>
> A4: Yes, this conclusion holds for ARM as well. For example, by finetuning ARM-B on semantic segmentation achieves 47.3 mIoU. This result is 2.0 higher than finetuning MambaMLP-B(45.3 mIoU) and 2.1 higher than finetuning Vim-B (45.2 mIoU). We are currently conducting experiments on object detection, which require more computational resources. We will share these results as soon as they are available; but if not being able to catch the discussion deadline, we will ensure they are incorporated in the final revision.
>
> Q5: Discuss with one related work.
>
> A5: Thank you for bringing this relevant work to our attention. Both works aim to pretrain models using autoregressive modeling techniques — while ARM pre-trains Mamba by predicting the next cluster, the method in [1] pre-trains CNN architectures by predicting surrounding masked regions. We will elaborate more on this discussion in the next version.
>
> [1] Chen, Y., Dai, X., Chen, D., Liu, M., Yuan, L., Liu, Z., & Lin, Y. (2023). Image as First-Order Norm+ Linear Autoregression: Unveiling Mathematical Invariance. arXiv preprint arXiv:2305.16319.

---

> > ### Comment · Reviewer_xhQF · 2024-11-26
> > **Thank you for the response**
> >
> > Thank you to the authors for the detailed reply. My concerns have been addressed, and I will be keeping my score. I appreciate you incorporating these changes into the final draft.

---

> > > ### Author Response · Authors · 2024-11-27
> > > **Thanks!**
> > >
> > > We are glad to see that all of your concerns have been addressed. Thank you once again for your support of our work.

---

### Meta-Review · Area_Chair_RK53 · 2024-12-22

**Metareview:**

The paper focuses on pre-training SSL strategies for MAMBA, showing that it is an effective strategy similar to ViT architectures. Most reviewers are positive about the paper, although there are some concerns regarding whether the comparisons with the best SSL competitors are sufficient. The authors acknowledge that the method is competitive but not necessarily better than the existing baselines, and they claim that what is important is to present an alternative architectural strategy compared to the standard ViT. I completely agree with this, thus I recommend acceptance.

**Additional Comments On Reviewer Discussion:**

The reviewers had differing views on the experimental evaluation of AR under Mamba in Vision. One perspective appreciates the demonstration of the method’s utility through pretraining on ImageNet-1k and applying it to variant datasets, with semantic segmentation results viewed as a supportive bonus, while acknowledging potential improvements with larger-scale experiments. Another perspective remains skeptical, citing a lack of strong novelty and evidence, noting that the results on ImageNet-1k and ADE20K are not competitive with benchmarks like the MAE paper, and emphasizing the need for additional evaluations on COCO object detection and segmentation to better establish the method’s effectiveness.

---

### Decision · Program_Chairs · 2025-01-22

Accept (Poster)